# Connecting Healthcare with Income Maximisation Services: A Systematic Review on the Health, Wellbeing and Financial Impacts for Families with Young Children

**DOI:** 10.3390/ijerph19116425

**Published:** 2022-05-25

**Authors:** Jade Burley, Nora Samir, Anna Price, Anneka Parker, Anna Zhu, Valsamma Eapen, Diana Contreras-Suarez, Natalie Schreurs, Kenny Lawson, Raghu Lingam, Rebekah Grace, Shanti Raman, Lynn Kemp, Rebecca Bishop, Sharon Goldfeld, Susan Woolfenden

**Affiliations:** 1Sydney Children’s Hospital Network, Sydney, NSW 2031, Australia; j.burley@unsw.edu.au (J.B.); Nora.samir97@gmail.com (N.S.); r.lingam@unsw.edu.au (R.L.); 2Population Child Health Research Group, School of Women and Children’s Health, University of NSW, Randwick, NSW 2031, Australia; anneka.parker@unsw.edu.au; 3BestSTART-SWS, Ingham Institute of Applied Medical Research, Liverpool, NSW 2170, Australia; v.eapen@unsw.edu.au (V.E.); rebekah.grace@westernsydney.edu.au (R.G.); 4Centre of Excellence for The Digital Child, The University of Wollongong, Wollongong, NSW 2522, Australia; 5Centre for Community Child Health, The Royal Children’s Hospital, Parkville, VIC 3052, Australia; anna.price@mcri.edu.au (A.P.); natalie.schreurs@mcri.edu.au (N.S.); sharon.goldfeld@rch.org.au (S.G.); 6Population Health, Murdoch Children’s Research Institute, Parkville, VIC 3052, Australia; 7Department of Paediatrics, The University of Melbourne, Parkville, VIC 3052, Australia; 8School of Economics, Marketing and Finance, RMIT University, Melbourne, VIC 3000, Australia; anna.zhu@rmit.edu.au; 9Academic Unit of Child Psychiatry, School of Psychiatry, Faculty of Medicine, University of New South Wales, Sydney, NSW 2170, Australia; 10Melbourne Institute, Applied Economic & Social Research, University of Melbourne, Parkville, VIC 3010, Australia; diana.contreras@unimelb.edu.au; 11Translational Health Research Institute (THRI), Western Sydney University, Penrith, NSW 2751, Australia; k.lawson@westernsydney.edu.au; 12Centre for the Transformation of Early Education and Child Health (TeEACH), Western Sydney University, Campbelltown, NSW 2560, Australia; lynn.kemp@westernsydney.edu.au; 13Community Paediatrics, South Western Sydney Local Health District, Liverpool, NSW 2170, Australia; shanti.raman@health.nsw.gov.au; 14Wesley Mission, 220 Pitt Street, Sydney, NSW 2500, Australia; rebeccam.bishop@wesleymission.org.au; 15Sydney Institute Women, Children and Their Families, Sydney Local Health District, Sydney, NSW 2050, Australia

**Keywords:** child health and wellbeing, poverty, income maximisation, public health, healthcare

## Abstract

Financial counselling and income-maximisation services have the potential to reduce financial hardship and its associated burdens on health and wellbeing in High Income Countries. However, referrals to financial counselling services are not systematically integrated into existing health service platforms, thus limiting our ability to identify and link families who might be experiencing financial hardship. Review evidence on this is scarce. The purpose of this study is to review “healthcare-income maximisation” models of care in high-income countries for families of children aged between 0 and 5 years experiencing financial difficulties, and their impacts on family finances and the health and wellbeing of parent(s)/caregiver(s) or children. A systematic review of the MEDLINE, EMBase, PsycInfo, CINAHL, ProQuest, Family & Society Studies Worldwide, Cochrane Library, and Informit Online databases was conducted according to the Preferred Reporting Items for Systemic Reviews and Meta-Analyses (PRISMA) statement. A total of six studies (five unique samples) met inclusion criteria, which reported a total of 11,603 families exposed to a healthcare-income maximisation model. An average annual gain per person of £1661 and £1919 was reported in two studies reporting one Scottish before–after study, whereby health visitors/midwives referred 4805 clients to money advice services. In another UK before–after study, financial counsellors were attached to urban primary healthcare centres and reported an average annual gain per person of £1058. The randomized controlled trial included in the review reported no evidence of impacts on financial or non-financial outcomes, or maternal health outcomes, but did observe small to moderate effects on child health and well-being. Small to moderate benefits were seen in areas relating to child health, preschool education, parenting, child abuse, and early behavioral adjustment. There was a high level of bias in most studies, and insufficient evidence to evaluate the effectiveness of healthcare-income maximisation models of care. Rigorous (RCT-level) studies with clear evaluations are needed to assess efficacy and effectiveness.

## 1. Introduction

It is recognised that social determinants of health (SDOH: conditions in which people are born, grow, live, work, and age) [1] directly or indirectly impact children’s health and development [2,3]. Childhood poverty detrimentally impacts every dimension of health, wellbeing and development [4]. As children raised in poverty become adults, they experience a greater subsequent risk of educational failure, underemployment, lack of societal participation, risky behaviours and delinquency [5]. It is estimated that one in five children in high-income countries live in relative income poverty [6], and experience severe multiple material deprivations that include (but not exclusively), low-quality housing, inadequate nutrition, and a lack of educational opportunities. These children are typically living in young, single-parent households, with an experience of disability and/or chronic disease, or families who do not speak English [7]. Compounding this, globally, there has been economic disruption by COVID-19, exacerbating and entrenching disadvantage due to widespread financial stress and job losses [8]. This has resulted in major adverse consequences on the health and wellbeing of families worldwide [9].

Internationally, researchers and policymakers have sought to design effective interventions that directly reduce child poverty and improve health equality [7]. Systematic reviews have reported unconditional cash transfers and increases in household cash income in low-, middle-, and high-income countries, which have been associated with beneficial effects on children’s health [10,11,12]. Although causal mechanisms underlying these effects are complex, the Investment Model [10] and the Family Stress Model [13] attempt to provide insights. The Investment Model suggests families invest additional economic resources into their children, addressing material deprivation and improving health and developmental opportunities [10], and the Family Stress Model suggests increasing economic resources can reduce parental stress and, in turn, increase emotionally responsive parenting and improve the home environment [13]. Broadly, it is understood that interventions to reduce household poverty will nurture children’s health and development.

Opportunities to address the impact of financial deprivation are also possible through service partnerships, whereby health and income-maximisation services work together. Services that provide advice on how to maximise income (i.e., financial counselling and Welfare advice services) are common in high-income countries, such as the United Kingdom and Australia. These services can increase awareness of financial entitlements for families, provide assistance with childcare, housing options, and debt management, and offer support and advice in crisis situations [14]. Linking families to existing organisations such as income maximisation services via healthcare offers the opportunity for wide-scale identification of financial hardship and may address the negative impacts on health and wellbeing [14]. However, there is a disparate field of evidence exploring the impact of financial services for families experiencing financial hardship, especially in the healthcare-income maximisation context, and this has yet to be reviewed systematically [15,16,17,18,19,20]. Such evidence is essential to develop a holistic service response to reduce family stress and invest in families to reduce the adverse impact of poverty on children’s health and wellbeing. To address the evidence gap, the purpose of this systematic review is to synthesise and critically evaluate the scientific evidence on the impact of healthcare—income maximisation models of care for families of children aged between 0 and 5 years experiencing financial difficulties in family finances and the health and wellbeing of parent(s)’/caregiver(s) or children. This paper presents the following: the study methodology in Section 2; a description of included studies and results in Section 3; a discussion and the implications of the results in Section 4, followed by a conclusion in Section 5.

## 2. Materials and Methods

### 2.1. Protocol and Registration

This systematic review was conducted following the Preferred Reporting Items for Systematic Reviews and Meta-Analysis (PRISMA-P) guidelines [21]. This review protocol is registered in the International Prospective Register of Systematic Reviews (PROSPERO) database and can be accessed at: https://www.crd.york.ac.uk/prospero/display_record.php?ID=CRD42020195985 (accessed on 27 April 2022) It has also been published [22]. See Appendix A for the completed PRISMA-P checklist.

### 2.2. Search Strategy

The strategy (Appendix A) followed the Population/Intervention/Comparison/Outcomes (PICO) approach [23] to ensure a systematic search of the literature. For key concepts of the population, search terms included subject headings (e.g., MeSH in PubMed/MEDLINE), in additionl to free text words, with suitable truncation. An experienced librarian helped develop the search strategy, which was then adapted for each bibliographic database. Article searches were conducted in specialised and general databases from the establishment date of databases to January 2021: Medical Literature Analysis and Retrieval System Online (MEDLINE), Excerpta Medica database (Embase), Psychology Information (PsycINFO), and (EMCare) via OVID. The Cumulative Index to Nursing and Allied Health Literature (CINAHL Complete), Proquest, and Family & Society Studies Worldwide via EBSCO. Cochrane Library via Wiley, and Informit Online via RMIT. Grey literature (e.g., book chapters, dissertations, conference abstracts, government reports/guidelines) were also searched. The reference sections of the included studies and cited studies were manually searched for additional relevant studies.

### 2.3. Eligibility Criteria

The eligibility criteria are published in detail in our protocol [22]. In brief, experimental and quasi-experimental study designs were eligible if they were conducted in high income countries with families of children aged between 0 and 5 years of age, who accessed a healthcare service that included a healthcare-income maximisation model of care. By way of definition, healthcare refers to services supporting families with young children such as paediatricians, child health nurses, general practitioners, health social workers, health professionals, nurses, doctors, or midwives. Income maximisation services refers to a breadth of services that are community-based and free and provide financial counselling, e.g., financial counsellors, citizens advice bureau, and financial literacy support. Income maximisation services may provide any of a range of supports, including examining family income, sources of income, current debts (utility, taxation, gambling, business, debt recovery, loans, or credit cards), unemployment, change in circumstances (e.g., disability), budgeting and expenses (accommodation, utilities, food, travel), and financial literacy/financial education. There was no minimum follow-up length, but studies had to measure at least one of the following outcomes: (1) income (change in income/earnings/debt management); (2) other financial impact (financial literacy); (3) parental/caregiver or child health and wellbeing, including physical, mental and spiritual health, social wellbeing, developmental wellbeing of the child, parenting skills, service use, cost of the intervention, and harm caused by the intervention. Studies were ineligible if they were conducted in low/middle income countries; focused on cash transfer programs; and/or used qualitative methodology or case series designs (<30 participants). To be included in this review, the paper had to be available in English.

### 2.4. Study Selection and Data Extraction

Bibliographic records were extracted from the database interfaces and imported into Reference Manager Software (Covidence, Melbourne, Australia) for de-duplication. Title and abstracts of potentially relevant studies retrieved from all sources (databases, web searches and citations of relevant studies) were screened by JB, AP and SW using Covidence software (a secure, internet-based software). Of these studies meeting initial eligibility screening, full texts were obtained, and screened by two reviewers. Discrepancies were resolved through discussion with a third reviewer. Using the format of the validated standard data extraction form, we extracted the following information: descriptive study characteristics (e.g., author, publication year, study design, country, sample size, age, sex), exposure, outcome, results, and confounders. Statistical significance was classified at *p* < 0.05. Reviewers were not blinded to the authors or journals when extracting data.

### 2.5. Assessment of Methodological Quality

Using the GRADE (Grading of Recommendations Assessment, Development and Evaluation) framework [24], we systematically examined the quality of research contributing to each outcome reflecting the level of confidence in the estimated effects. The assessment criteria (risk of bias, inconsistency, indirectness, imprecision, other) were used to rate quality of evidence as “high”, “moderate”, “low”, or “very low”. There is no official tool when assessing risk of bias in observational studies using the GRADE framework, which is one component of the quality of evidence. For non-randomised studies, we used the Risk Of Bias In Non-randomized Studies of Interventions (ROBINS-I) assessment tool developed by Cochrane [25]. Study quality did not influence eligibility for inclusion.

### 2.6. Data Synthesis

The limited number of studies and heterogeneity in their settings, intervention delivery, and outcomes impeded meta-analysis. The results of this systematic review are therefore presented narratively.

## 3. Results

Figure 1 presents the PRISMA flow diagram. The search strategy produced 6061 studies. Based on titles and abstracts, 47 full text studies were reviewed, of which 6 (five unique samples) met the eligibility criteria and were critically appraised. Table 1 summarises the participants, interventions and outcomes for the six included studies. Reasons for excluding the 42 ineligible studies are outlined in Figure 1.

### 3.1. Study Designs and Participants

Study designs included two RCTs (one unique sample) conducted in New Zealand [26,27] and four before and after studies [28,29,30,31] in Scotland, UK and USA. Study participants ranged from 107 families [30] to 6248 families [31], with a total of 11,603 families exposed to a healthcare-income maximisation model of care. Of the six eligible studies, only the RCT included a control group of 223 families [26,27]. The race/ethnicity of participants was reported in four of six publications; one targeted minority ethnic groups, and in the other three, minority ethnicity groups made up a third and a quarter of participants [26,27]. All studies targeted families of children. Of these, four of six studies reported the age range; Reading [30] included families of children less than 1 year of age, Reading [30] & Fergusson [26,27] included families of children between 0 and 5 years of age Fergusson [26,27] and Parthasarathy [31] included families of children aged between 1 and 5.

### 3.2. Description of Interventions

Three of the six studies reported interventions where nurses/family support workers provided financial support during home visits. The first of these was part of the Early Start Program in New Zealand, which primarily focused on child health, preschool education and parenting outcomes and were linked to financial support, although this was not clarified [26,27]. The second article reported the Building Economic Security Today (BEST) program in USA, which aimed to help families maximise their income and increase their financial assets through the home visitation programme [31]. Public Health Nurses (PHNs) distributed a questionnaire during the first home visit that assessed: (1) financial impact of having a child born with medical concerns, (2) adequacy of income, (3) financial difficulty and strain, (4) knowledge of Supplemental Security Income (SSI), and (5) perceived financial stress. PHNs also asked families to identify financial topics they wanted to learn more about and offered resources to the family based on the primary caregiver’s responses. A resource packet was also provided [31].

Two of the six studies reported interventions of health visitor/midwife referring clients to money advice services. These studies were before and after studies of the Healthier Wealthier Children (HWC) project in Scotland, conducted with different participants and timepoints [28,29]. HWC is an NHS-led child poverty initiative aimed at tackling child poverty across NHS Greater Glasgow and Clyde (NHSGGC). Referral links were established between health and financial advice services. Development officers and money advice workers established communication links with frontline health staff, raised awareness about the project, addressed training needs and supported staff to refer clients on to local advice services. HWC developed four different types of service delivery: (1) ‘core’ service offered by local money advice services (scheduled/drop-in appointment system); (2) outreach at various Community Health (and Care) Partnership (CH(C)P) locations; (3) home-based appointments; and (4) telephone appointments. The service delivery methods offered in each area were dependent upon the location of HWC advice services, availability of outreach space, home visit policies and the capacity of advice services.

Of the six studies reviewed, one described financial counsellors offering advice to eligible families with a child less than 1 year of age in an urban primary health care centre in the UK [30]. Families were approached by mail and by health visitors who encouraged family participation in the study.

### 3.3. Quality Assessment

Using GRADE, risk of bias of included studies was independently assessed by NS and SW. There was a high level of agreement in the rating of studies between the authors. Table 2 presents the risk of bias for the 5 publications. Red indicates a high risk, green indicates a low risk, and yellow indicates an unclear risk of bias.

A randomisation process was reported in a single randomised controlled trial of two unique studies [26,27]. The other four publications described before–after study designs with no randomisation [28,29,30,31] All studies reported high risk of bias.

### 3.4. Outcomes

#### 3.4.1. Financial Impact

Financial impacts were reported in four of the six included studies and are summarised in Table 3. Financial gain was the most commonly assessed outcome among financial outcomes and was included in all of the four studies. In the pre- and post-studies, there was an average annual gain per person who had the intervention of £1661 [28], £1919 [29], across ten communities across Glasgow, indicating a large population scale impact, and, £1058 [30]. In the one randomised controlled trial there were no significant differences in financial gain outcomes and other financial impacts; however, families randomised to the intervention fared better than the control group; for welfare dependence (70.1% vs. 71.5%) and reported a higher average income at 36 months (NS$499 vs. NZ$492), respectively [26,27]—Table 3.

#### 3.4.2. Non-Financial Impact

Engagement was assessed in two pre- and post-studies. In the first study 2516 referrals were made between health and financial advice services and advice uptake was reported for 1346 families [28]. In the second study, wherein 2289 referrals were made between health and financial advice services, advice uptake was reported for 1027 families [29]. Other non-financial gains included advice on benefit entitlements and timing of eligibility, childcare, employment and housing tenancy issues and payment to creditors [29], reported in Table 4. No economic evaluation or potential intervention harm were described in any studies.

#### 3.4.3. Parental/Child Health and Well Being

Parental and child health was assessed in two studies (one unique sample) [26,27]. There was a lack of association observed between the intervention group, compared to the control group, for outcomes of maternal health, specifically depression and substance abuse, *p* > 0.05 [27]; reported in Table 5. For children enrolled in the intervention, small to moderate effects were observed for childhood functioning that include: higher rates of contact with doctors, frequently attended well-child checks and dental services, fewer hospital attendances (*p* < 0.05, effects sized ranged from 0.03 to 0.25 respectively), higher enrolment rates at preschool and service use of local and welfare agencies (*p* < 0.01, effect size ranged from 0.22 to 0.31), higher rates of positive parenting and non-punitive parenting (*p* < 0.05, effect size ranged from 0.22 to 0.26), lower rates of child assault (*p* < 0.1, effect size 0.26), and improvements in behaviour at 36 month assessment (*p* < 0.5, effect sized ranged from 0.19 to 0.26; Table 6).

## 4. Discussion

This systematic review is the first study to synthesise the evidence for healthcare-income maximisation models of care for families of children aged 0–5 years who are experiencing financial difficulties, and to examine the impacts on family finances and health and wellbeing of parent/caregiver(s) and children. From the six studies included in this review, a total of 11,603 families were exposed to a healthcare-income maximisation model of care intervention. There was a high level of bias in most studies, with only two RCTs (one unique sample) and four pre- and post-studies. The outcome with the most evidence of impact was financial gain. The two studies describing the Scottish HWC pre- post study (HWC), where health visitors/midwives referred clients to money advice services, reported an average annual gain per person of £1661 [28] and £1919 [29]. Another UK pre- and post-study, whereby financial counsellors were attached to urban primary healthcare centres, reported an average annual gain per person of £1058 [30]. This preliminary evidence suggests that healthcare-income maximisation models can help address financial hardship experienced by families with young children, and is similar to evidence indicating that income maximisation programmes tackling the SDOH delivered through healthcare services can help with financial benefits for families with low income [32]. However, the single randomized controlled trial did not report financial gain resulting from the Early Start Program [27].

Increasing referrals to support services may also be viewed as a positive impact. The HWC programme generated 5003 referrals to advice services, and provided advice on entitlements and timing of eligibility, childcare, employment, housing tenancy issues and payment to creditors [29]. This finding is consistent with broader evidence on the positive impact of systematic approaches to identifying and addressing unmet social needs [33]. Two studies also reported positive impacts on financial literacy, increasing parental agency and confidence in managing finances in the future [29,31]. Through embedding income maximisation models of care, we are addressing a key SDOH, that is poverty and financial stress. By asking families about their social needs for the SDOH we can connect families with multiple support resources that are beyond that of direct health services provided. These include support and advocacy for childcare, employment, and/or housing that can help to mitigate childhood poverty and equity globally [34].

Only two studies (one unique sample) reported on parents/caregivers and/or child health and wellbeing outcomes [26,27]. Despite clear benefits being observed for the Early Start service, the evaluation showed positive changes in parenting- and child-related outcomes [27]; yet, no significant changes in maternal health or family level changes [26]. The observed child outcomes from Fergusson et al. [27] are similar to a RCT conducted by Olds et al. [35], that examined the long-term effects of a nurse visitation program of prenatal and early childhood over 15 years. This has important implications for policy to show that the provision of support, advice and mentoring to help child welfare, such as building parenting knowledge and skills, can help children thrive.

Overall, the scientific evidence was limited and heterogeneous, and the quality of evidence was low. Only five studies reporting four interventions across three high-income countries (UK, Scotland, NZ) were included. In one before–after study [30], the intervention’s short duration prevented it becoming sufficiently established in primary care and generating positive impact. Additionally, it focussed on parents of infants, which the authors argued may have been too limited, given that financial difficulties often increase as the child grows out of infancy [30]. Additional limitations of this review were the heterogeneity in the reporting and measurement of outcomes. Non-English language electronic databases and publications were excluded.

### Implications for Future Research & Service Delivery

Family income and financial security are essential for children to achieve optimal health and development [4]. Social support to increase family income and help protect them from poverty, such as cash transfers, income support payments, and community services, are used in many high-income countries. As such, services that aim to maximise income have great potential for supporting families experiencing, or at-risk of, poverty. It is evident from the included studies that financial gain was more prominent in families that had access to an income-maximisation model. Further increased access of services and addressing unmet social needs, which has been shown to mitigate the impact of childhood poverty [33], were evident. Systematically embedding and evaluating these practices into routine healthcare may offer a novel avenue for directly addressing financial hardship and its impacts on families’ health and wellbeing at scale, and also provide opportunities for health care and not-for profit services to collaborate to improve the health and well-being of families. This lack of robust evidence and evaluation regarding their effectiveness indicates the need for methodologically rigorous future trials examining these efforts. As we continue to respond to, and recover from, the COVID pandemic, healthcare alone will not address social determinants of health, such as poverty. To support families in need through early intervention, we need to design and test holistic models of care that include addressing unmet social needs, that have the potential for far-reaching effects for families and service delivery stakeholders, including healthcare professionals, financial counsellors and money advice workers, health policy managers and researchers, to provide a holistic service response. Studies should also consider measuring the effect size of how embedding these services into healthcare may be associated with a gain or loss of health, so as to measure the severity of impact at an individual or population level.

## 5. Conclusions

Linking healthcare and income maximisation services shows a potentially promising way to address childhood poverty and increase the effectiveness and efficiency of existing infrastructure. While a small number of studies suggest that these models can generate financial gain for families with young children, there is insufficient evidence to date to indicate their efficacy. Studies that address a key SDOH, such as poverty, using rigorous evaluation of healthcare-income maximisation models of care, and that explore their impacts on financial, health and wellbeing outcomes for families with young children, are urgently required in response to childhood poverty and equity globally.

## Figures and Tables

**Figure 1 ijerph-19-06425-f001:**
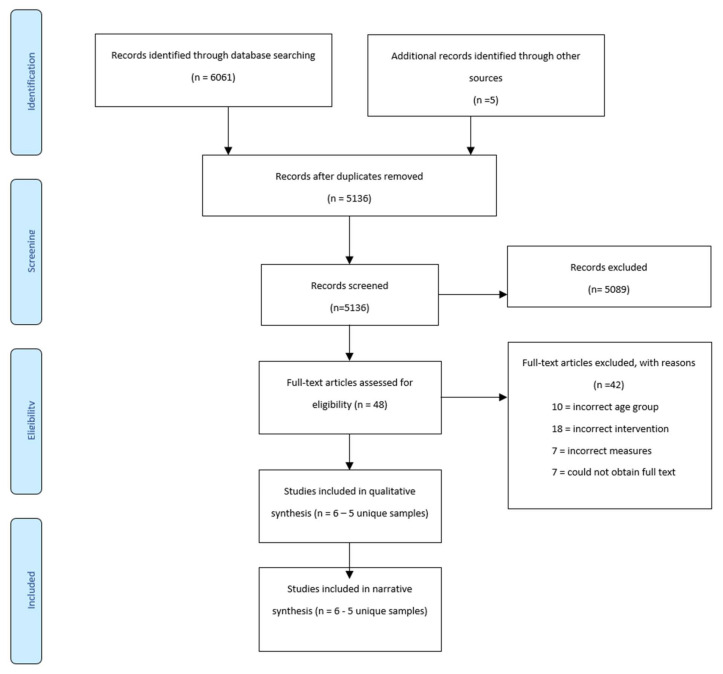
Prisma Diagram of search results [21].

**Table 1 ijerph-19-06425-t001:** Description of Included Publications.

	Paper	Study Type	Setting	Participants	Demographics	Intervention, Control Description	Outcomes Measures
1	Fergusson 2006 [26] (New Zealand)	Randomised controlled trial	Families enrolled in the Christchurch urban region by community nurses	*n* = 443 families I = 220 families C = 223 families Attrition: Intervention: 16.4%; Control: 7.2% Participants identified by community nurses who screened families with 2 or more risk factors	Single parent family (64.6% intervention vs. 63.8% in control. Families predominantly welfare dependent, with low income, and had parents with limited educational achievement. Although the client population consisted of predominantly white New Zealanders, the rate of Māori (the indigenous people of New Zealand) parents was approximately twice that of the rate of Māori in the general New Zealand population. *	*Nurses/family support workers providing financial support during home visits* Family support workers (FSWs) who had nursing or social work qualifications visited families at home as part of the Early Start Program and encouraged family economic and material wellbeing: reducing levels of welfare dependence, encouraging the use of budgeting services, encouraging workforce participation, and encouraging forward economic planning. Control group randomised from trial recruitment; No Early Start Program; Control group received $50 (New Zealand) per interview for their time	Welfare dependence, family weekly income, mother in paid employment, partner in paid employment, number of economic hardship factors
2	Fergusson 2005 [27] (New Zealand)	Randomised controlled trial	Families enrolled in the Christchurch urban region by community nurses	*n* = 443 families I = 220 families C = 223 families Attrition: Intervention: 16.4%; Control: 7.2% Participants identified by community nurses who screened families with 2 or more risk factors	Single parent family (64.6% intervention vs. 63.8% in control. Families predominantly welfare dependent, with low income, and had parents with limited educational achievement. Although the client population consisted of predominantly white New Zealanders, the rate of Māori (the indigenous people of New Zealand) parents was approximately twice that of the rate of Māori in the general New Zealand population. *	*Nurses/family support workers providing financial support during home visits* Family support workers (FSWs) who had nursing or social work qualifications visited families at home as part of the Early Start Program and encouraged family economic and material wellbeing: reducing levels of welfare dependence, encouraging the use of budgeting services, encouraging workforce participation, and encouraging forward economic planning. Control group randomised from trial recruitment; No Early Start Program; Control group received $50 (New Zealand) per interview for their time	Medical outcomes (visits to doctor; immunisations; wellbeing checks; hospital attendance for accidents/injuries; if child was enrolled in dental services). Use of preschool education and welfare utilisation; parenting practices; child abuse & neglect; child behaviour.
3	Naven 2012 [28] (Scotland)	Before and after study	Ten Community Health (and Care) Partnership (CH(C)P) areas that existed across NHS Greater Glasgow and Clyde (NHS GGC)	*n* = 2516 referrals Overall uptake of advice services = 54%	Lone parents; 59% Minority ethnic groups in south and west Glasgow successfully reached	*Health visitor/midwife referring clients to money advice services* Healthier, Wealthier Children (HWC) project; a range of early years staff (e.g., health visitor/midwife) referred eligible clients to local HWC money advice services. Local HWC services contacted client and offered advice, intervention and onward referral if required.	Financial Gain: Annual gain (£); Number (%) gain cases; Average gain per case (£) Engagement: Referrals; Advice uptake
4	Naven 2013 [29] (Scotland)	Before and after study	Ten Community Health (and Care) Partnership (CH(C)P) areas that existed across NHS Greater Glasgow and Clyde (NHS GGC)	*n* = 2289 referrals Overall uptake of advice services = 45%	Lone parents = 69% (703/1012); Couples = 31% (318/1012) White (67%; 818/1213), Black or Minority Ethnic background (BME) 12% (146/1213)	*Health visitor/midwife referring clients to money advice services* Healthier, Wealthier Children (HWC) project; a range of early years staff (e.g., health visitor/midwife) referred eligible clients to local HWC money advice services. Local HWC services contacted client and offered advice, intervention and onward referral if required.	Financial Gain: Annual gain (£); Number (%) gain cases; Average gain per case (£) Engagement: Referrals; Advice uptake
5	Reading 2001 [30] (UK)	Before and after study	Three urban primary health care centres in Norwich, UK	*n* = 107 families Overall uptake of advice service = 23 (22%)	Lone parents = 24% (21/87)	*Financial counsellors attached to urban primary health care centres.* A trained Citizens Advice Bureau worker (financial advisor and other social service support) was attached to three urban primary health care centres in Norwich, UK, for 1 day per week over a period of 9 months.	Welfare benefits, debt, legal, housing, utilities, taxation, employment, consumer rights, relationships
6	Parthasarathy 2003 [31] (USA)	Before and after study	Women, Infants, and Children Program (WIC) client families for BEST financial educational classes; Medically Vulnerable Infant Program (MVIP) for BEST financial assessments during public health nurse home visits.	BEST financial educational classes: N = 6248 WIC client families; 1592 (26%) completed post-class participant surveys BEST MVIP home visitation program: N = 163 infants Primary caregivers of 139 (85%) infants completed BEST questionnaires.	WIC client families, all living at the federal poverty level or below; infants at risk for neurological problems and developmental delays because of prematurity, low birth weight or other medical conditions experienced at birth, and discharged neonatally from a California Children’s Services approved neonatal intensive care unit.	*Nurses/family support workers providing financial support during home visits* One-on-one support to families (public health nurses) in home visiting programs; financial education classes for Women, Infants and Children Program (WIC) clients; asset development educational materials and referrals for all clients	Understanding of the health-wealth connection; knowledge of asset development strategies and resources; confidence and readiness to improve financial behaviours; improved financial behaviours; stress levels

* In all comparisons of demographics, there were no significant differences between the intervention and control groups. I = Intervention group, C = Control.

**Table 2 ijerph-19-06425-t002:** Risk of Bias Summary—based on GRADE [24].

Study	Random Sequence Generation	Allocation Concealment	Blinding of Outcome Assessment	Incomplete Outcome Data	Selective Reporting	Other Reporting Bias
Fergusson 2005 [28]	Low	High	High	High	Low	Low
Fergusson 2006 [27]	Low	High	High	High	Low	Low
Naven 2012 [29]	High	High	Unclear	High	Low	Low
Naven 2013 [30]	High	High	Unclear	High	Low	Low
Reading 2001 [31]	High	High	Unclear	High	Low	Low
Parthasarathy 2003 [32]	High	High	Unclear	High	Low	Unclear

**Table 3 ijerph-19-06425-t003:** Financial Impact.

Paper	Outcome Measure	Results	Significance
* **Financial gain** *			
Fergusson 2006 [26]	Welfare dependent at 36 mo, %	70.1 (intervention) vs. 71.5 (control)	*p* = 0.76
	Family weekly income at 36 mo, mean, New Zealand Dollars	499 (intervention) vs. 492 (control)	*p* = 0.64
Naven 2012 [28]	No. of financial gain cases	663 (49%)	Not reported
	Benefits and savings One-off payments Total	£2,030,915 £225,807 £2,256,722	Not reported
	Average annual gain per person	£1661	Not reported
Naven 2013 [29]	Annual total and debt managed One-off payments Total	£1,941,533 £35,147 £ 1,976,680	Not reported
	Average annual gain per person	£1919	Not reported
Reading 2001 [30]	One-off payments Annual recurring payments Total	£17,857 £6480 £24,337	Not reported
	Average annual gain per person	£1058	Not reported
* **Other financial impacts** *			
Fergusson 2006 [26]	Mother in paid employment at 36 mo, %	31.5 (intervention) vs. 26.6 (control)	0.28
	With partner in paid employment at 36 mo, %	27.2 (intervention) vs. 30.4 (control)	0.48
	No. of hardship factors (past 12 mo), mean	4.5 (intervention) vs. 4.2 (control)	0.32

**Table 4 ijerph-19-06425-t004:** Non-Financial Impact.

Paper	Outcome Measure	Results	Significance
* **Engagement** *			
Naven 2012 [28]	Referrals	2516	Not reported
	Advice uptake	1346 (54%)	Not reported
Naven 2013 [29]	Referrals	2289	Not reported
	Advice uptake	1027 (45%)	Not reported
	Onward referral to services	N = 110 (8%)	Not reported
* **Financial literacy** *			
Naven 2013 [29]	Qualitative	Clients appear to feel more confident in managing their money.	N/A
Parthasarathy 2014 [31]	Qualitative and Survey	Increased clients’ awareness of financial issues.	N/A

**Table 5 ijerph-19-06425-t005:** Parental health and wellbeing.

Paper	Outcome Measure	Results	Significance
	Maternal Depression (past 12 mo), % ^a^	16.9 (intervention) vs. 16.9 (control)	0.81
Fergusson 2006 [26]	Maternal Substance Use: smoked cigarettes (per day) at 36 mo, %	62.0 (intervention) vs. 62.3 (control)	0.94
	Alcohol use problems (past 12 mo), %	14.1 (intervention) vs. 9.7 (control)	0.17
	Substance use problems (past 12 mo), %	4.9 (intervention) vs. 5.8 (control)	0.69

^a^ Composite International Diagnostic Interview scale were used to determine whether parents met the Diagnostic and Statistical Manual of Mental Disorders, Fourth Edition.

**Table 6 ijerph-19-06425-t006:** Child health and wellbeing.

Paper	Measure	Results (Intervention vs. Control)	Association 95% CI	Significance
Fergusson 2005 [27]	**Child Health**			
	Mean no. of GP visits (0–36 mo)	23.4 (intervention) vs. 20.7 (control)	0.11 (0.01–0.21)	<0.05
	% Up to date with immunisations (0–36 mo)	92.5 (intervention) vs. 91.9 (control)	1.09 (0.51–2.32)	0.83
	% Up to date with well-child checks (0–36 mo)	41.9 (intervention) vs. 30.1 (control)	1.70 (1.11–2.59)	<0.05
	% Attended hospital for accident/injury or accidental poisoning (0–36 mo)	17.5 (intervention) vs. 26.3 (control)	0.59 (0.36–0.98)	<0.05
	% Enrolled with dental nurse/dentist at 36 mo	72.3 (intervention) vs. 62.8 (control)	1.54 (1.01–2.37)	<0.05
	**Service utilization**			
	Mean duration of early childhood education, mo (0–36 mo)	16.4 (intervention) vs. 13.6 (control)	0.11 (0.01–0.21)	<0.05
	Mean no. of community service contacts (0–36 mo)	8.7 (intervention) vs. 7.7 (control)	0.16 (0.06–0.26)	<0.01
	**Maternal parenting attitudes ^†^**			
	Mean positive parenting attitudes (36 mo)	10.14 (intervention) vs. 9.88 (control)	0.13 (0.03–0.23)	<0.01
	Mean nonpunitive attitudes (36 mo)	10.12 (intervention) vs. 9.90 (control)	0.11 (0.01–0.21)	<0.05
	Mean parenting score (36 mo)	10.14 (intervention) vs. 9.87 (control)	0.13 (0.03–0.23)	<0.01
	**Child abuse and neglect**			
	% Parental report of severe physical assault (0–36 mo)	4.4 (intervention) vs. 11.7 (control)	0.35 (0.15–0.80)	<0.01
	% In contact with agencies for child abuse or neglect (0–36 mo)	19.6 (intervention) vs. 21.3 (control)	0.91 (0.55–1.48)	0.39
	**Child behavioral adjustment ^†^**			
	Mean externalising score (36 mo)	9.90 (intervention) vs. 10.09 (control)	0.09 (−0.01–0.19)	<0.07
	Mean internalising score (36 mo)	9.86 (intervention) vs. 10.12 (control)	0.13 (0.03–0.23)	<0.01
	Mean total behavior score (36 mo)	9.87 (intervention) vs. 10.11 (control)	0.12 (0.02–0.22)	<0.05

**^†^** All parenting and child behavior scores were standardized to a mean of 10 and an SD of 1.

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
