# Peer review of "Connecting Healthcare with Income Maximisation Services: A Systematic Review on the Health, Wellbeing and Financial Impacts for Families with Young Children"

_ijerph, 2022, doi:10.3390/ijerph19116425_

Round 1
Reviewer 1 Report
Please, see the attached document

Author Response
1. Despite that families randomized to the intervention fared better than the control group; for welfare dependence and reported a higher average income, the one RCT shows no significant differences in financial gain. How could this be explained?
- Reply: Thank you for your comments. The primary focus of this Intervention was not financial gain, but rather parenting as part of parenting support package that was linked to financial services. This point was addressed in the original manuscript -outlined below:
- “However, the single randomized controlled trial did not report financial gain resulting from the Early Start Program (Fergusson, Grant et al. 2006). This may be because the Early Start Program did not specifically target family finances but rather prioritised home visiting and family support to optimize a range of other child health and parenting outcomes. In addition this result could also be confounded by the fact that the programme wanted to decrease welfare dependency and probably at this stage what made more sense was for families to access all the possible support that they could.”
- In agreeance with comments made by Reviewer one during the first round of revisions, we feel that the discussion should focus on financial impact, and not based on variables not contemplated.
2. An important finding was an effect in positive parenting and non-punitive parenting, this mean child wellbeing. Please include a short sentence about this finding on child welfare in the discussion
What do the new findings imply?
- Reply: The following has been added to the discussion.
- Despite clear benefits being observed for the Early Start service, the evaluation showed positive changes in parenting- and child-related outcomes [27],
- “This has important implications for policy to show that the provision of support, advice and mentoring to help child welfare; such as building parenting knowledge and skills can help children thrive.”
3. Line 64: Please correct, NS or NZ
- Reply: Thank you for point it. This has been amended to NZ.
Reviewer 2 Report
The authors improved the previous version.
In this version, the authors need to reference more of the main topics and review the English grammar and style.
Author Response
Thank you for your suggestions. The manuscript has been reviewed for grammar and revised throughout.
This manuscript is a resubmission of an earlier submission. The following is a list of the peer review reports and author responses from that submission.
Round 1
Reviewer 1 Report
The authors did an excellent work, but I had some very specific observations:
The aim is evaluating the scientific evidence on the impact of healthcare - income maximisation models of care for families of children aged 0-5 years experiencing financial difficulties on family finances and parent/caregiver(s) or children’s health and wellbeing; but is necessary to define who are the families and their health care needs, to find the scope of the proposed interventions…
On the other hand, the impact should be a gain or loss of health status. This means scale/severity of the impact (on mortality, morbidity and welfare) and size/proportion of the population affected (total, intermediate, low). In this sense, maybe, the authors should be use impact indicators to elaborate their health outcomes...
Therefore, in the Fergusson study, there was no statistical significance, which indicate that the impact may be due to other variables...
Obviously, there is a financial gain if there a financial support for the families, but the idea must be the gain in children´s health
Lines 288-294: Regardless of any other intention of the analyzed studies, the authors should discuss the observed results based on the financial impact and no discuss based on variables not contemplated...
Lines 313-321: What impact on children´s health?
Please, clarify, and discuss on results of this systematic review
Line 332-338: In my opinion, the authors should draw the effects on social determinants on child health outcomes, and from that, integrate the effect on the intention (income maximisation services).
Author Response
Please see the attachment - both reviewers comments are addressed in the attached word document.

Reviewer 2 Report
The paper is good, but the authors need to improve it. My comments are:
- Abstract: reinforce the paper's contribution; it is not clear the gap and the actual contribution.
- Introduction: not necessary www.financialcounsellingaustralia,org.au; only reference it. Close the introduction with a resume of the paper. For example, this paper presents.......... section 2. xxxxxx section 3.
- Materials and methods: improve this section. For the extension and the content, it is unnecessary to structure with: 2.1, 2.2. .... Explain and describe this section more.
- Results: Improve table 2. The authors could add more references or actual references and try to connect the contributions, Not only resume and describe the references how the authors connect it.
- Discussion: it is a standard discussion, may reconsider the structure, similar section 3. The content doesn´t justify subsections 4.1 4.2. The paper needs more numerical and statistical analysis to support this section.
- Conclusions: it is the poor section. Align it with the introduction and describe more the actual contribution of the paper. Try to write a qualitative and quantitative analysis.
Author Response

(The authors gave the same response as above.)
